# Sparse Weight Activation Training

**Md Aamir Raihan, Tor M. Aamodt**
Department of Electrical And Computer Engineering
University of British Columbia
Vancouver, BC
{araihan,aamodt}@ece.ubc.ca

## Abstract

Neural network training is computationally and memory intensive. Sparse training can reduce the burden on emerging hardware platforms designed to accelerate sparse computations, but it can also affect network convergence. In this work, we propose a novel CNN training algorithm called Sparse Weight Activation Training (SWAT). SWAT is more computation and memory-efficient than conventional training. SWAT modifies back-propagation based on the empirical insight that convergence during training tends to be robust to the elimination of (i) small magnitude weights during the forward pass and (ii) both small magnitude weights and activations during the backward pass. We evaluate SWAT on recent CNN architectures such as ResNet, VGG, DenseNet and WideResNet using CIFAR-10, CIFAR-100 and ImageNet datasets. For ResNet-50 on ImageNet SWAT reduces total floating-point operations (FLOPs) during training by 80% resulting in a $3.3\times$ training speedup when run on a simulated sparse learning accelerator representative of emerging platforms while incurring only 1.63% reduction in validation accuracy. Moreover, SWAT reduces memory footprint during the backward pass by 23% to 50% for activations and 50% to 90% for weights. Code is available at `https://github.com/AamirRaihan/SWAT`.

## 1 Introduction

Convolutional Neural Networks (CNNs) are effective at many complex computer vision tasks including object recognition [27, 57], object detection [56, 50] and image restoration [12, 65]. However, training CNNs requires significant computation and memory resources. Software and hardware approaches have been proposed for addressing this challenge. On the hardware side, graphics processor units (GPUs) are now typically used for training [27] and recent GPUs from NVIDIA include specialized Tensor Core hardware specifically to accelerate deep learning [46, 44, 45]. Specialized programmable hardware is being deployed in datacenters by companies such as Google and Microsoft [25, 7]. Techniques for reducing computation and memory consumption on existing hardware include those reducing the number of training iterations such as batch normalization [24] and enhanced optimization strategies [26, 13] and those reducing computations per iteration. Examples of the latter, which may be effective with appropriate hardware support, include techniques such as quantization [67, 6, 62, 59], use of fixed-point instead of floating-point [63, 8], sparsification [60] and dimensionality reduction [34]. This paper introduces *Sparse Weight Activation Training* (SWAT), which significantly extends the sparsification approach.

Training involves repeated application of forward and backward passes. Prior research on introducing sparsity during training has focused on sparsifying the backward pass. While model compression [18, 40, 36, 35, 32, 21, 37, 61, 41] introduces sparsification into the forward pass, it typically does so by introducing additional training phases which increase overall training time. Amdahl's Law [2] implies overall speedup is limited by the fraction of original execution time spent on computations that are

not sped up by system changes. To reduce training time significantly by reducing computations per training iteration it is necessary to address both forward and backward passes. SWAT introduces sparsification into both forward and backward passes and is suitable for emerging hardware platforms containing support for sparse matrix operations. Such hardware is now available. For example recently announced Ampere GPU architecture [46] includes support for exploiting sparsity. In addition, there is a growing body of research on hardware accelerators for sparse networks [47, 9, 66, 1] and we demonstrate, via hardware simulation, that SWAT can potentially train $5.9\times$ faster when such accelerators become available.

While SWAT employs sparsity it does so with the objective of *reducing training time* **not** *performing model compression*. The contributions of this paper are:

- An empirical sensitivity analysis of approaches to inducing sparsity in network training;
- SWAT, a training algorithm that introduces sparsity in weights and activations resulting in reduced execution time in both forward and backward passes of training;
- An empirical evaluation showing SWAT is effective on complex models and datasets.

## 2    Related work

Below we summarize the works most closely related to SWAT.

**Network pruning:**    LeCun et al. [31] proposed removing network parameters using second-order information of the loss function to improve generalization and reduce training and inference time. More recently, pruning has been explored primarily as a way to improve the efficiency and storage requirements of inference but at the expense of increasing training time, contrary to the objective of this paper. Specifically, Han et al. [18] showed how to substantially reduce network parameters while improving validation accuracy by pruning weights based upon their magnitude combined with a subsequent retraining phase that fine-tunes the remaining weights. Regularization techniques can be employed to learn a pruned network [40, 36]. Various approaches to structured pruning [61, 41, 32, 35, 21, 37], ensure entire channels or filters are removed to reduce inference execution time on the vector hardware found in GPUs.

**Reducing per-iteration training overhead:**    MeProp [55, 60] reduces computations during training by back propagating only the highest magnitude gradient component and setting other component to zero. As shown in Section 3.2, on complex networks and models training is very sensitive to the fraction of gradient components set to zero. Liu et al. [34] proposed reducing computation during training and inference by constructing a dynamic sparse graph (DSG) using random projection. DSG incurs accuracy loss of $3\%$ on ImageNet at a sparsity of $50\%$. Goli and Aamodt [17] propose to reduce backward pass computations by performing convolutions only on gradient components that change substantially from the prior iteration. They reduced the overall computation in the backward pass by up to 90% with minimum loss in accuracy. Their approach of backward pass sparsification is orthogonal to SWAT.

**Sparse Learning:**    Sparse learning attempts to learn a sparse representation during training, generally as a way to achieve model compression and reduce computation during inference. Since sparse learning introduces sparsity during training it can potentially reduce training time but pruning weights by itself does not reduce weight gradient computation (Equation 3). Many sparse learning algorithms start with a random sparse network, then repeat a cycle of training, pruning and regrowth. Sparse Evolutionary Training (SET) [39] prunes the most negative and smallest positive weights then randomly selects latent (i.e., missing) weights for regrowth. Dynamic Sparse Reparameterization (DSR) [42] uses a global adaptive threshold for pruning and randomly regrows latent weights in a layer proportionally to the number of active (non-zero) weights in that same layer. Sparse Network From Scratch (SNFS) [10] further improves performance using magnitude-based pruning and momentum for determining the regrowth across layers. Rigging the Lottery Ticket (RigL) [15] uses an instantaneous gradient as its regrowth criteria. Dynamic Sparse Training (DST) [33] defines a trainable mask to determine which weights to prune. Recently Kusupati et al. [30] proposes a novel state-of-the-art method of finding per layer learnable threshold which reduces the FLOPs during inference by employing a non-unform sparsity budget across layers.

Table 1: Comparison of SWAT with related works

| Algorithms | Sparsity Across Layer | Sparse Forward Pass | Sparse Backward Pass | | Unstructured Sparse Network | Structured Sparse Network |
|---|---|---|---|---|---|---|
| | | | Input Gradient | Weight Gradient | | |
| Network Pruning | Fixed/Variable | Yes/Gradual | Yes/Gradual | No | Depend on algorithm | |
| meProp [55] | Fixed | No | Yes | Yes | No | - |
| DSG [34] | Fixed | Yes | Yes | Yes | Yes | - |
| SET [39] | Fixed | Yes | Yes | No | Yes | - |
| DSR [42] | Variable | Yes | Yes | No | Yes | - |
| SNFS [10] | Variable | Yes | Yes | No | Yes | - |
| RigL [15] | Fixed | Yes | Yes | No | Yes | - |
| SWAT | Fixed/Variable | Yes | Yes | Yes | Yes | Yes |

In contrast, SWAT employs a unified training phase where the algorithm continuously explores sparse topologies during training by employing simple magnitude based thresholds to determine which weight and activations components to operate upon. In addition, the sparsifying function used in SWAT can be adapted to induce structured sparse topologies and varying sparsity across layers. Table 1 summarizes the differences between SWAT and recent related work on increasing sparsity. The column "Sparsity Across Layer" indicates whether the layer sparsity is constant during training. The columns labeled "Input Gradient" and "Weight Gradient" represent whether the input gradient or weight gradient computation is sparse during training.

## 3 Sparse weight activation training

We begin with preliminaries, describe a sensitivity study motivating SWAT then describe SWAT and several enhancements.

### 3.1 Preliminaries

We consider a CNN trained using mini-batch stochastic gradient descent, where the $l^{th}$ layer maps input activations $a_{l-1}$ to outputs $a_l$ using function $f_l$:

$$a_l = f_l(a_{l-1}, w_l) \tag{1}$$

where $w_l$ are layer $l$'s weights. During back-propagation the $l^{th}$ layer receives the gradient of the loss with respect to its output activation ($\bigtriangledown_{a_l}$). This is used to compute the gradient of the loss with respect to its input activation ($\bigtriangledown_{a_{l-1}}$) and weight ($\bigtriangledown_{w_l}$) using function $G_l$ and $H_l$ respectively. Thus, the backward pass for the $l^{th}$ layer can be defined as:

$$\bigtriangledown_{a_{l-1}} = G_l(\bigtriangledown_{a_l}, w_l), \tag{2}$$
$$\bigtriangledown_{w_l} = H_l(\bigtriangledown_{a_l}, a_{l-1}) \tag{3}$$

We induce sparsity on a tensor by retaining the values for the $K$ highest magnitude elements and setting the remaining elements to zero. Building on the notion of a "Top-k" query in databases [52] and simlar to Sun et al. [55] we call this process Top-$K$ sparsification.

### 3.2 Sensitivity Analysis

We begin by studying the sensitivity of network convergence by applying Top-$K$ sparsification to weights ($w_l$), activations ($a_{l-1}$) and/or back-propagated error gradients ($\bigtriangledown_{a_l}$) during training. For these experiments, which help motivate SWAT, we evaluate DenseNet-121, VGG-16 and ResNet-18 on the CIFAR-100 dataset. We run each experiment three times and report the mean value.

Figure 1 plots the impact on validation accuracy of applying varying degrees of Top-$K$ sparsification to weights or activations during computations in the forward pass (Equation 1). In this experiment, when applying top-$K$ sparsification in the forward pass weights or activations are not permanently

removed. Rather, low magnitude weights or activations are removed temporarily during the forward pass and restored before the backward pass. Thus, in this experiment the backward pass uses unmodified weights and activations without applying any sparsification and all weights are updated by the resulting dense weight gradient. The data in Figure 1 suggests convergence is more robust to Top-$K$ sparsification of weights versus activations during the forward pass. The data shows that inducing high activation sparsity hurts accuracy after a certain point, confirming similar observations by Georgiadis [16] and Kurtz et al. [29].

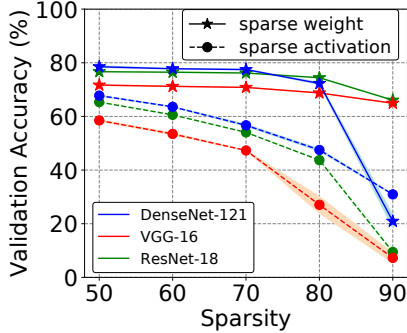 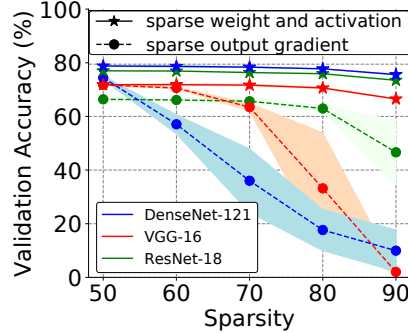

Figure 1: Forward Pass Sensitivity Analysis.  Figure 2: Backward Pass Sensitivity Analysis.

Figure 2 plots the impact on validation accuracy of applying varying degrees of Top-$K$ sparsification to both weights and activations (labeled "sparse weight and activation") or to only output gradients (labeled "sparse output gradient") during computations in the backward pass (Equations 2 and 3). "Weights and activations" or "back-propagated output error gradients" are sparsified before performing the convolutions to generate weight and input gradients. The generated gradients are dense since a convolution between a sparse and dense inputs will in general produce a dense output. The resulting dense weight gradients are used in the parameter update stage. We note that this process differs, for example, from recent approaches to sparsifying inference (e.g., [30, 68, 15]) which employ dense back-propagated output gradients during convolution to generate weight gradients that are masked during the parameter update stage. The "sparse weight and activation" curve shows that convergence is relatively insensitive to applying Top-$K$ sparsification. In contrast, the "sparse output gradient" curve shows that convergence is sensitive to applying Top-$K$ sparsification to back-propagated error gradients ($\bigtriangledown_{a_l}$). The latter observation indicates that meProp, which drops back-propagated error-gradients, will suffer convergence issues on larger networks.

## 3.3 The SWAT Algorithm

The analysis above suggests two strategies: In the forward pass use sparse weights (but *not* activations) and in the backward pass use sparse weights and activations (but *not* gradients).

Sparse weight activation training (SWAT) embodies these two strategies as follows (for pseudo-code see supplementary material): During each training pass (forward and backward iteration on a mini-batch) a sparse weight topology is induced during the forward pass using the Top-$K$ function. This partitions weights into active (i.e., Top-K) and non-active sets for the current iteration. The forward pass uses the active weights. In the backward pass, the full gradients and active weights are used in Equation 2 and the full gradients and highly activated neurons (Top-$K$ sparsified activations) are used in Equation 3. The later generate dense weight gradients that are used to update *both* active and non-active weights. The updates to non-active weights mean the topology can change from iteration to iteration. This enables SWAT to perform dynamic topology exploration: Backpropagation with sparse weights and activations approximates backpropagation on a network with sparse connectivity and sparsely activated neurons. The dense gradients generated during back-propagation minimize loss for the current sparse connectivity. However, the updated weights resulting from the dense gradients will potentially lead to a new sparse network since non-active weights are also updated. This process captures fine-grained temporal importance of connectivity during training. Section 4.5 shows quantitatively, the importance of unmasked gradient updates and dynamic exploration of connectivity.

### 3.3.1 Top-K Channel Selection

Top-$K$ sparsification induces fine-grained sparsity. SWAT can instead induce structured sparsity on weights by pruning channels. Similarly to [4, 35, 38, 32], the saliency criteria for selecting channels is $L_1$ norm. Figure 3 illustrates using channel $L_1$ norm to select 50% of channel (Top-50%). The squares on the right side contain the channel $L_1$ norm and lower $L_1$ norm channels are set as non-active (lightly shaded). The importance of channels is consider independently, i.e., different filters can select different active channels.

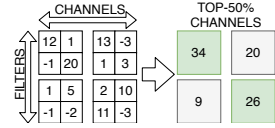

Figure 3: Top-K Channel Selection.

### 3.3.2 Sparsity Distribution

The objective of SWAT is to reduce training time while maintaining accuracy. So far we have assumed the per layer sparsity of weights and activations is equal to the target sparsity. Prior work [15, 10] has demonstrated that a non-uniform distribution of active weights and activation sparsity can improve accuracy for a given sparsity. We explore three strategies to distributing sparsity. All three of the following variants employ magnitude comparisons to select which tensor elements to set to zero to induce sparsity. We say an element of a tensor is unmasked if it is not forced to zero before being used in a computation. For all three techniques below the fraction of unmasked elements in a given layer is identical for weights and activations.

**Uniform (SWAT-U):** Similar to others (e.g., Evci et al. [15]) we found that keeping first layer dense improves validation accuracy. For SWAT-U we keep the first layer dense and apply the same sparsity threshold uniformly across all other layers.

**Erdos-Renyi-Kernel (SWAT-ERK):** For SWAT-ERK active weights and unmasked activations are distributed across layers by taking into account layer dimensions and setting a per layer threshold for magnitude based pruning. Following Evci et al. [15] higher sparsity is allocated to layers containing more parameters.

**Momentum (SWAT-M):** For SWAT-M active weights and unmasked activations are distributed across layers such that less non-zero elements are retained in layers with smaller average momentum for active weights. This approach is inspired by Dettmers and Zettlemoyer [10].

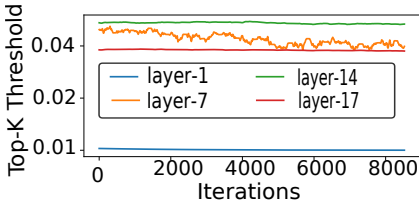

Figure 4: Top-$K$ weight threshold versus training iteration

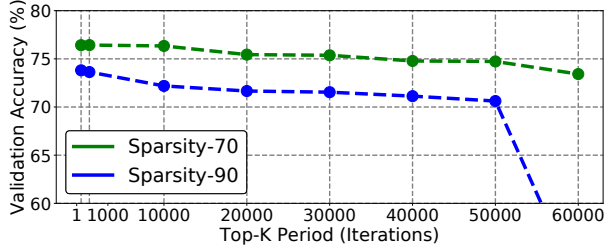

Figure 5: Impact of Top-$K$ sampling period.

### 3.3.3 Efficient Top-K Threshold Calculation

The variants of SWAT described above induce sparsity by examining the magnitude of tensor elements and this incurs some overhead. Naively, the Top-$K$ operation could be performed on a 1-dimensional array of size $N$ in $O(N \log N)$ using sorting. The overhead can be reduced to $O(N)$ using a threshold operation where elements with magnitude greater than a threshold are retained an others are treated as zeros. The $K$-th largest element can be found in $O(N)$ average time complexity using Quickselect [22] or in $\theta(N)$ time using either BFPRT [3] or Introselect [43] and efficient parallel implementations of Top-$K$ have been proposed for GPUs [52].

Figure 4 plots the threshold value required to achieve 90% weight sparsity versus training iteration for SWAT-U and unstructured pruning four representative layers of ResNet-18 while training using

CIFAR-100. This data suggests that for a given layer the magnitude of the $K$-th largest element is almost constant during training. Thus, we explored whether we can set a per layer threshold for weights and activations on the first iteration and only update it periodically.

Figure 5 plots the final Top-1 validation accuracy after training converges versus Top-$K$ sampling period for SWAT-U and unstructured pruning applied to ResNet-18 trained using CIFAR 100. Here 391 iterations (x-axis) corresponds to a single epoch. The data indicates that for sampling intervals up to 1000 iterations validation accuracy is not degraded significantly.

## 4   Experiments

Below we present results for validation accuracy, theoretical reduction in floating-point operations (FLOPs) during training and estimates of training speedup on a simulated sparse accelerator. While **not** our primary objective we also report theoretical FLOPs reduction for inference.

### 4.1   Methodology

We measure validation accuracy of SWAT by implementing custom convolution and linear layers in PyTorch 1.1.0 [48]. Inside each custom PyTorch layer we perform sparsification before performing the layer forward or backward pass computation. To obtain accuracy measurements in a reasonable time these custom layers invoke NVIDIA's cuDNN library using Pytorch's C++ interface.

We estimate *potential* for training time reduction using an analytical model to measure total floating-point operations and an architecture simulator modeling a sparse DNN training accelerator based upon an extension of the Bit-Tactical inference accelerator [9].

We employ standard training schedules for training ResNet [19], VGG [53] with batch-normalization [24], DenseNet [23] and Wide Residual Network [64] on CIFAR10 and CIFAR100 [28]. We use SGD with momentum as an optimization algorithm with an initial learning rate of $0.1$, momentum of $0.9$, and weight decay $\lambda$ of $0.0005$. For training runs with ImageNet we employ the augmentation technique proposed by Krizhevsky et al. [27]: $224 \times 224$ random crops from the input images or their horizontal flip are used for training. Networks are trained with label smoothing [58] of $0.1$ for 90 epochs with a batch size of 256 samples on a system with eight NVIDIA 2080Ti GPUs. The learning rate schedule starts with a linear warm-up reaching its maximum of $0.1$ at epoch 5 and is reduced by $(1/10)$ at epochs $30^{th}$, $60^{th}$ and $80^{th}$. The optimization method is SGD with Nesterov momentum of $0.9$ and weight decay $\lambda$ of $0.0001$. Results for ImageNet use a Top-$K$ threshold recomputed every 1000 iterations while those for CIFAR-10 recompute the Top-$K$ threshold every iteration. Due to time and resource constraints below we report SWAT-ERK and SWAT-M results only for CIFAR-10 and unstructured sparsity.

The supplementary material includes detailed hyperparameters, results for CIFAR-100, ablation study of the accuracy impact of performing Top-$K$ on different subsets of weight and activation tensors.

### 4.2   Unstructured SWAT

**CIFAR-10:**   Table 2 compares SWAT-U, SWAT-ERK and SWAT-M with unstructured sparsity versus published results for DST [33] and SNFS [10] for VGG-16-D, WRN-16-8, and DenseNet-121 on CIFAR-10. Under the heading "Training Sparsity" we plot the average sparsity for weights (W) and activations (Act). For SWAT-ERK and SWAT-M per layer sparsity for weights and activations are equal for a given layer but their averages for the entire network differ because these are computed by weighting by the number of weights or activations per layer. Comparing SWAT against SNFS and DST for VGG-16 and WRN-16-8 the data shows that SWAT-M has better accuracy versus SNFS and DST on both networks. While SWAT-M reduces training FLOPs by 33% and 22% versus SNFS and DST, respectively, SWAT-M requires more training FLOPs versus DST on VGG-16. SWAT-U has better accuracy versus SNFS and DST on WRN-16-8 and SWAT-U obtains $2.53\times$ and $1.96\times$ harmonic mean reduction in remaining training FLOPS versus SNFS and DST, respectively. It is important to note that the reduction in FLOPs for SWAT-ERK is competitive with SWAT-U. In general, uniform will require fewer training and inference computations verses ERK when the input resolution is high since ERK generally applies lower sparsity at the initial layer resulting in significant initial overhead. However, in Table 2, for all these networks the initial layer will have less computation

Table 2: Unstructured SWAT on the CIFAR-10 dataset. **W:** Weight, **Act:** Activation, **BA:** Baseline Accuracy, **AC:** Accuracy Change, **MC:** Model Compression, **DS:** Default Sparsity.

| Network | Methods | Training Sparsity | | Top-1 | | Training | | Inference | |
|---|---|---|---|---|---|---|---|---|---|
| | | W (%) | Act (%) | Acc. ± SD (σ) (%) | BA / AC | **FLOPs↓** (%) | Act↓ (%) | MC (×) | FLOPs↓ (%) |
| VGG-16 | SNFS [10] | 95.0 | DS | 93.31 | 93.41 / -0.10 | 57.0 | DS | 20.0 | 62.9 |
| | DST [33] | 96.2 | DS | 93.02 | 93.75 / -0.73 | 75.7 | DS | 26.3 | 83.2 |
| | SWAT-U | 90.0 | 90.0 | 91.95 ± 0.06 | 93.30 / -1.35 | 89.7 | 36.2 | 10.0 | 89.5 |
| | SWAT-ERK | 95.0 | 82.0 | 92.50 ± 0.07 | 93.30 / -0.80 | 89.5 | 33.0 | 20.0 | 89.5 |
| | SWAT-M | 95.0 | 65.0 | **93.41 ± 0.05** | 93.30 / **+0.11** | 64.0 | 25.0 | 20.0 | 58.4 |
| WRN-16-8 | SNFS [10] | 95.0 | DS | 94.38 | 95.43 / -1.05 | 81.8 | DS | 20.0 | 88.0 |
| | DST [33] | 95.4 | DS | 94.73 | 95.18 / -0.45 | 83.3 | DS | 21.7 | 91.4 |
| | SWAT-U | 90.0 | 90.0 | **95.13 ± 0.11** | 95.10 / **+0.03** | 90.0 | 49.0 | 10.0 | 90.0 |
| | SWAT-ERK | 95.0 | 84.0 | 95.00 ± 0.12 | 95.10 / -0.10 | 91.4 | 45.8 | 20.0 | 91.7 |
| | SWAT-M | 95.0 | 78.0 | 94.97 ± 0.04 | 95.10 / -0.13 | 86.3 | 42.5 | 20.0 | 85.9 |
| DenseNet-121 | SWAT-U | 90.0 | 90.0 | **94.48 ± 0.06** | 94.46 / **+0.02** | 89.8 | 44.0 | 10.0 | 89.8 |
| | SWAT-ERK | 90.0 | 88.0 | 94.14 ± 0.11 | 94.46 / -0.32 | 89.7 | 43.0 | 10.0 | 89.6 |
| | SWAT-M | 90.0 | 86.0 | 94.29 ± 0.11 | 94.46 / -0.17 | 84.2 | 42.0 | 10.0 | 83.6 |

Table 3: Unstructured SWAT on the ImageNet dataset. **W:** Weight, **Act:** Activation, **BA:** Baseline Accuracy , **AC:** Accuracy Change, **MC:** Model Compression, **DS:** Default Sparsity.

| Network | Methods | Training Sparsity | | Top-1 | | Training | | Inference | |
|---|---|---|---|---|---|---|---|---|---|
| | | W (%) | Act (%) | Acc. ± SD (σ) (%) | BA / AC | **FLOPs↓** (%) | Act↓ (%) | MC (×) | FLOPs↓ (%) |
| ResNet-50 | SET [39] | 80.0 | DS | 73.4 ±0.32 | 76.8 / -3.4 | 58.1 | DS | 3.4 | 73.0 |
| | | 90.0 | DS | 71.3 ±0.24 | 76.8 / -5.5 | 63.8 | DS | 5.0 | 82.1 |
| | DSR [42] | 80.0 | DS | 74.1 ±0.17 | 76.8 / -2.7 | 51.6 | DS | 3.4 | 59.4 |
| | | 90.0 | DS | 71.9 ±0.07 | 76.8 / -4.9 | 58.9 | DS | 5.0 | 70.7 |
| | SNFS [10] | 80.0 | DS | 74.9 ±0.07 | 77.0 / -2.1 | 45.8 | DS | 5.0 | 43.3 |
| | | 90.0 | DS | 72.9 ±0.07 | 77.0 / -4.1 | 57.6 | DS | 10.0 | 59.7 |
| | RigL [15] | 80.0 | DS | 74.6 ±0.06 | 76.8 / -2.2 | 67.2 | DS | 5.0 | 77.7 |
| | | 90.0 | DS | 72.0 ±0.05 | 76.8 / -4.8 | 74.1 | DS | 10.0 | 87.4 |
| | DST [33] | 80.4 | DS | 74.0 ±0.41 | 76.8 / -2.8 | 67.1 | DS | 5.0 | 84.9 |
| | | 90.1 | DS | 72.8 ±0.27 | 76.8 / -4.0 | 75.8 | DS | 10.0 | 91.3 |
| | SWAT-U | **80.0** | **80.0** | **75.2 ±0.06** | 76.8 / **-1.6** | **76.1** | **39.2** | **5.0** | 77.7 |
| | | 90.0 | 90.0 | 72.1 ±0.03 | 76.8 / -4.7 | 85.6 | 44.0 | 10.0 | 87.4 |
| | SWAT-ERK | **80.0** | **52.0** | **76.0 ±0.16** | 76.8 / **-0.8** | **60.0** | **25.5** | **5.0** | 58.9 |
| | | 90.0 | 64.0 | 73.8 ±0.23 | 76.8 / -3.0 | 79.0 | 31.4 | 10.0 | 77.8 |
| | SWAT-M | **80.0** | **49.0** | **74.6 ±0.11** | 76.8 / **-2.2** | **45.9** | **23.7** | **5.0** | 45.0 |
| | | 90.0 | 57.0 | 74.0 ±0.18 | 76.8 / -2.8 | 65.4 | 27.2 | 10.0 | 64.8 |
| WRN-50-2 | SWAT-U | 80.0 | 80.0 | 76.4 ±0.10 | 78.5 / -2.1 | 78.6 | 39.1 | 5.0 | 79.2 |
| | | 90.0 | 90.0 | 74.7 ±0.27 | 78.5 / -3.8 | 88.4 | 43.9 | 10.0 | 89.0 |

due to the small input resolution of the CIFAR-10 dataset, and computationally expensive layers are allotted higher sparsity in SWAT-ERK.

The data under "Act ↓" report the fraction of activation elements masked to zero, which can be exploited by hardware compression proposed by NVIDIA [51] to reduce transfer time between GPU and CPU.

**ImageNet:** Table 3 compares SWAT-U with unstructured sparsity against six recently proposed sparse learning algorithms at target weight sparsities of 80% and 90% on ImageNet. Data for all sparse learning algorithms except RigL were obtained by running their code using the hyperparameters in Section 4.1. Results for RigL are quoted from Evci et al. [15]. SET and DSR do not sparsify downsampling layers leading to increased parameter count. At 80% sparsity, SWAT-U attains the highest validation accuracy while reducing training FLOPs. DST trains ResNet-50 on ImageNet dataset with less computation ("Training FLOPs") than RigL even though DST is a dense to sparse training algorithm while RigL is a sparse to sparse training algorithm. This occurs for two reasons: (1) DST quickly reaches a relatively sparse topology after a few initial epochs; in our experiment, DST discards more

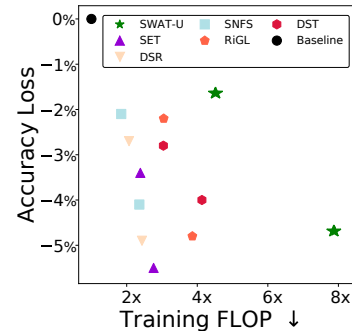

Figure 6: Accuracy decrease versus reduction in training FLOPs

Table 4: Structured SWAT on the CIFAR-10 dataset. **W:** Weight, **Act:** Activation, **CP:** Channel Pruned, **BA:** Baseline Accuracy, **AC:** Accuracy Change, **MC:** Model Compression.

| Network | Methods | Training Sparsity | | Top-1 | | Training | | | Inference |
| | | W (%) / Act (%) | CP (%) | Acc. ± SD (σ) (%) | BA / AC | FLOPs↓ (%) | Act↓ (%) | MC (×) | FLOPs↓ (%) |
|---|---|---|---|---|---|---|---|---|---|
| ResNet-18 | SWAT-U | 50.0/50.0 | 50.0 | 94.73±0.06 | 94.59 / +0.14 | 49.8 | 26.0 | 2.0 | 49.8 |
| | | 60.0/60.0 | 60.0 | 94.68±0.03 | 94.59 / +0.09 | 59.8 | 31.2 | 2.5 | 59.8 |
| | | 70.0/70.0 | 70.0 | 94.65±0.19 | 94.59 / +0.06 | 69.8 | 36.4 | 3.3 | 69.8 |
| DenseNet-121 | SWAT-U | 50.0/50.0 | 50.0 | 95.04±0.26 | 94.51 / +0.53 | 49.9 | 25.0 | 2.0 | 49.9 |
| | | 60.0/60.0 | 60.0 | 94.82±0.11 | 94.51 / +0.31 | 59.9 | 30.0 | 2.5 | 59.9 |
| | | 70.0/70.0 | 70.0 | 94.81±0.20 | 94.51 / +0.30 | 69.9 | 35.0 | 3.3 | 69.9 |

Table 5: Structured SWAT on the ImageNet dataset. **W:** Weight, **Act:** Activation, **CP:** Channel Pruned, **BA:** Baseline Accuracy, **AC:** Accuracy Change, **MC:** Model Compression.

| Network | Methods | Training | | | Pruning | | Top-1 | | Inference |
| | | W (%) / Act (%) | FLOPs↓ (%) | Act↓ (%) | CP (%) | Fine-Tune (epoch) | Acc (%) | BA / AC | FLOPs↓ (%) |
|---|---|---|---|---|---|---|---|---|---|
| ResNet-50 | DCP [69] | Offline Pruning | | | - | 60 | 74.95 | 76.01 / -1.06 | 55.0 |
| | CCP [49] | Offline Pruning | | | 35 | 100 | 75.50 | 76.15 / -0.65 | 48.8 |
| | AOFP [11] | Offline Pruning | | | - | Yes (-) | 75.11 | 75.34 / -0.23 | 56.73 |
| | Soft-Pruning [20] | - | - | - | 30 | No | 74.61 | 76.15 / -1.54 | 41.8 |
| | SWAT-U (Structured) | 50.0/50.0 | 47.6 | 24.5 | 50 | No | 76.51±0.30 | 76.80 / -0.29 | 48.6 |
| | | 60.0/60.0 | 57.1 | 29.5 | 60 | No | 76.35±0.06 | 76.80 / -0.45 | 58.3 |
| | | 70.0/70.0 | 66.6 | 34.3 | 70 | No | 75.67±0.06 | 76.80 / -1.13 | 68.0 |
| WRN-50-2 | SWAT-U (Structured) | 50.0/50.0 | 49.1 | 24.5 | 50 | No | 78.08±0.20 | 78.50 / -0.42 | 49.5 |
| | | 60.0/60.0 | 58.9 | 29.4 | 60 | No | 77.55±0.07 | 78.50 / -0.95 | 59.4 |
| | | 70.0/70.0 | 68.7 | 34.2 | 70 | No | 77.19±0.11 | 78.50 / -1.31 | 69.3 |

than 70% of network parameter within 5 epochs; (2) The sparsity distribution across layers is the crucial factor deciding reduction in FLOPs since allocating higher sparsity to the computationally expensive layer alleviates the initial overhead during entire training. Therefore, the overall benefit of DST is dependent on network architecture, sparsity distribution, and the parameter discard rate. Figure 6 plots the reduction in validation accuracy versus the reduction in training FLOPs relative to baseline dense training at both 80% and 90% sparsity using the same data as Table 3. This figure shows SWAT-U provides the best tradeoff between validation and reduction in training FLOPs.

### 4.3 Structured SWAT

**CIFAR10:** Table 4 provides results for SWAT-U with channel pruning on the CIFAR-10 dataset. At 70% sparsity SWAT-U with channel pruning improves validation accuracy on both ResNet-18 and DenseNet-121 while reducing training FLOPs by $3.3\times$.

**ImageNet:** Table 5 compares SWAT-U with channel pruning against four recent structured pruning algorithms. At 70% sparsity SWAT-U with channel pruning reduces training FLOPs by $3.19\times$ by pruning 70% of the channels on ResNet50 while incurring only $1.2\%$ loss in validation accuracy. SWAT-U with structured sparsity shows better accuracy but shows larger drops versus our baseline (trained with label smoothing) in some cases. The works we compare with start with a densely trained network, prune channels, then fine-tune and so increase training time contrary to our objective.

### 4.4 Sparse Accelerator Speedup

We believe SWAT is well suited for emerging sparse machine learning accelerator hardware designs like NVIDIA's recently announced Ampere (A100) GPU. To estimate the speedup on a sparse accelerator, we have used an architecture simulator developed for a recent sparse accelerator hardware proposal [9]. The simulator counts only the effectual computation (non-zero computation) and exploits the sparsity present in the computation. The architecture has a 2D array of processing units where each processing unit has an array of multipliers and dedicated weight and activation and accumulation buffers. It counts the cycles taken to spatially map and schedules the computation present in each layer of the network. The memory hierarchy is similar to the DaDianNo architecture [5]. The activation and weight buffer can hold one entire layer at a time and hide the latency of memory transfer. The memory throughput is high enough to satisfy the computation throughput.

Table 6: Speedup due to SWAT on ML Accelerator

| Top-$K$ Sparsity | Forward Pass Speed Up | Backward Pass Speed Up |
|:---:|:---:|:---:|
| 0% | 1× | 1× |
| 80% | 3.3× | 3.4× |
| 90% | 5.6× | 6.3× |

The simulator only implements the forward pass, and therefore the simulator does not simulate the backward pass. However, the backward pass convolution is a transposed convolution, which can be translated into a standard convolution by rotating the input tensor [14]. So we estimated the backward pass speedup by transforming transpose convolution into a standard convolution and using the inference simulator. Note, here the transformation overhead was not considered in the speedup. However, the overhead would be small in the actual hardware since this transformation operation is a simple rotation operation, i.e., rotating the tensor along some axis and, therefore, we assume it could be accelerated on hardware. Table 6 reports forward and backward pass training speedup (in simulated clock cycles) for SWAT-U with unstructured pruning on ResNet-50 with ImageNet. From Table 3 we see that at 80% sparsity SWAT-U incurs 1.63% accuracy loss but the data in Table 6 suggests it may improve training time by 3.3× on emerging platforms supporting supporting hardware acceleration of sparse computations.

### 4.5 Effect of updates to non-active weights

SWAT updates non-active weights in the backward pass using dense gradients and one might reasonable ask whether it would be possible to further decrease training time by masking these gradients. Figure 7 measures the effect of this masking on validation accuracy while training ResNet-18 on CIFAR-100. The data suggest it is important to update non-active weights. Since non-active weights are updated they may become active changing the topology. Figure 8 shows the effect of freezing the topology after differing number of epochs during training of ResNet18 on CIFAR100 dataset. Freezing topology exploration early is harmful to convergence and results in reduced final validation accuracy.

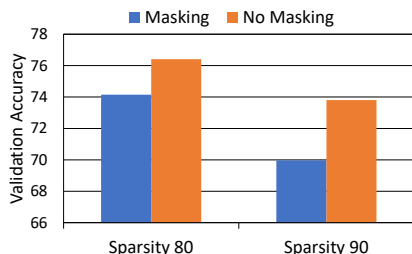

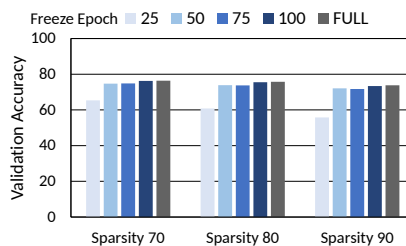

Figure 7: Effect of masking of gradient update for non-active weights.

Figure 8: Effect of stopping dynamic topology exploration early.

## 5 Conclusion

In this work, we propose SWAT, a novel efficient training algorithm that sparsifies both the forward and the backward passes with negligible impact on convergence. SWAT is the first sparse learning algorithm we are aware of that can train both structured and unstructured sparse networks. SWAT achieves validation accuracy comparable to other pruning and sparse learning algorithms while demonstrating potential for significant reduction in training time on emerging platforms supporting hardware acceleration of sparse computations.

## 6 Broader Impact

This work has the following potential positive impact in society: 1) It is an entirely sparse training algorithm for emerging sparse hardware accelerators. 2) It makes training efficient and faster. Thus,

decreasing the model training cost. 3) It can reduce the overall carbon footprint of model training, which is a huge problem. Strubell et al. [54] shows that the carbon footprint of training models can release $5\times$ the carbon emission of a car during its lifetime. 4) It can enable us to train even bigger models, thus allowing us to achieve new state-of-the-art accuracies.

At the same time, this work may have some negative consequences because SWAT can enable us to develop better AI and better AI technologies may have negative societal implications such as strict surveillance, privacy concerns, and job loss.

## 7   Acknowledgements

We thank Aayush Ankit, Francois Demoullin, Negar Goli, Dave Evans, Deval Shah, Yuan Hsi Chou, and the anonymous reviewers for their valuable comments on this work. This research has been funded in part by the Computing Hardware for Emerging Intelligent Sensory Applications (COHESA) project. COHESA is financed under the National Sciences and Engineering Research Council of Canada (NSERC) Strategic Networks grant number NETGP485577-15. Professor Aamodt serves as a consultant for Huawei Technologies Canada.

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
