[Supplementary Material 1 · supplementary.pdf]

# Appendix A   Appendix

## A.1   Detailed description of the SWAT algorithm

The SWAT algorithm is summarized in Algorithm 1. The shaded region represents the sparse computation step which could be exploited on sparse machine learning accelerators. The SWAT algorithm consists of three parts: Forward Computation, Backward Computation, and the Parameter Update.

---

**Algorithm 1** SWAT Algorithm

---

**The data:** Training iteration $t$ and Top-$K$ sampling period $P$. Network with $L$ layers and previous weight parameters $\mathbf{w}^t$. Sparsity distribution algorithm $D$. Mini-batch of inputs ($\mathbf{a}_0$) and corresponding targets ($\mathbf{a}^*$). Learning rate $\eta$. Gradient descent optimization algorithm $Optimizer$.

**The result:** Updated weight parameters $\mathbf{w}^{t+1}$.

*Stage 1. Forward Computation*
1  **for** $l = 1$ to $L$ **do**
2     **if** $l$ *is a convolutional or fully-connected layer* **then**
3        **if** $t \bmod P = 0$ **then**
4           $S_l \Leftarrow \texttt{getLayerSparsity}(l, D)$
5           $t_l^w \Leftarrow \texttt{getThreshold}(\mathbf{w}_l, S_l, t)$
6           $t_l^a \Leftarrow \texttt{getThreshold}(\mathbf{a}_{l-1}, S_l, t)$
7        **end**
8        $\mathbf{W}_l^t \Leftarrow f_{TOPK}(\mathbf{w}_l^t, t_l^w)$
9        $\mathbf{a}_l^t \Leftarrow \texttt{forward}(\mathbf{W}_l^t, \mathbf{a}_{l-1}^t)$
10       $\mathbf{a}_{l-1}^t \Leftarrow f_{TOPK}(\mathbf{a}_{l-1}^t, t_l^a)$
11       **save_for_backward**$_l \Leftarrow \mathbf{W}_l^t, \mathbf{a}_{l-1}$
12    **else**
13       $\mathbf{a}_l \Leftarrow \texttt{forward}(\mathbf{w}_l^t, \mathbf{a}_{l-1})$
14       **save_for_backward**$_l \Leftarrow \mathbf{w}_l^t, \mathbf{a}_{l-1}$
15    **end**
16 **end**

*Stage 2. Backward Computation*
17 Compute the gradient of the output layer $\bigtriangledown_{\mathbf{a}_L} = \frac{\partial loss(\mathbf{a}_L, \mathbf{a}^*)}{\partial \mathbf{a}_L}$
18 **for** $l = L$ to $1$ **do**
19    $\mathbf{W}_l^t, \mathbf{a}_{l-1} \Leftarrow$ **save_for_backward**$_l$
20    **if** $l$ *is a convolutional or fully-connected layer* **then**
21       $\bigtriangledown_{\mathbf{a}_{l-1}} \Leftarrow \texttt{backward\_input}(\bigtriangledown_{\mathbf{a}_l}, \mathbf{W}_l^t)$
22       $\bigtriangledown_{\mathbf{w}_{l-1}} \Leftarrow \texttt{backward\_weight}(\bigtriangledown_{\mathbf{a}_l}, \mathbf{a}_{l-1}^t)$
23    **else**
24       $\bigtriangledown_{\mathbf{a}_{l-1}} \Leftarrow \texttt{backward\_input}(\bigtriangledown_{\mathbf{a}_l}, \mathbf{W}_l^t)$
25       $\bigtriangledown_{\mathbf{w}_{l-1}} \Leftarrow \texttt{backward\_weight}(\bigtriangledown_{\mathbf{a}_l}, \mathbf{a}_{l-1}^t)$
26    **end**
27 **end**

*Stage 3. Parameter Update*
28 **for** $l = 1$ to $L$ **do**
29    $\mathbf{w}_l^{t+1} \Leftarrow Optimizer(\mathbf{w}_l^t, \bigtriangledown_{\mathbf{w}_l}, \eta)$
30 **end**

---

The forward computation (Line 1 to 16) for each layer proceeds as follows: First, we check if the current layer is a convolutional or fully-connected layer (Line 2). If neither, perform the regular (non-SWAT) forward pass computation (Line 13) and save dense weights and activations (Line 14). Otherwise, if the training iteration $t$ is a multiple of the Top-$K$ sampling period $P$ then we obtain

the target sparsity $\mathbf{S}_l$ of layer $l$ based upon distribution algorithm $D$ (Line 4) where $D$ is one of the techniques described in Section 3.3.2. Then, we compute threshold $t^w$ for weight sparsity (Line 5) used in the forward pass and threshold $t^a$ for use in sparsifying activations prior to saving to memory for use in the the backward pass (Line 6). Prior to performing the forward computation (Line 9) we compute the active weights $\mathbf{W}_l^t$ by applying the sparsifying function, $f_{TOPK}$ using threshold $t^w$. For input tensor $\mathbf{x}_i$ For fine-grained sparsity $f_{TOPK}(\mathbf{x},t)$ maps input elements $x_i \in \mathbf{x}$ to output elements $y_i$ according to:

$$y_i = \begin{cases} 0, & |x_i| \leq t \\ x_i, & \text{otherwise} \end{cases}$$

For coarse-grained sparsity (Section 3.3.1), which we apply only to weights, $f_{TOPK}(\mathbf{x},t)$ maps input elements $x_i \in \mathbf{x}$ to output elements $y_i$ where $i$ is an element of channel $C$ according to:

$$y_i = \begin{cases} 0, & \sum_{i \in C} |x_i| \leq t \\ x_i, & \text{otherwise} \end{cases}$$

Next, we perform sparse forward computations, `forward`, corresponding to Equation 1 in the paper to generate output activation (Line 9). Next, we apply fine-grained Top-$K$ sparsification to the input activations (Line 10). Save sparse active weight parameters $\mathbf{W}_l^t$ and input activations $\mathbf{a}_{l-1}$ for the backward pass (Line 11).

After the forward pass the loss function is applied and the back propagated error-gradient for the output layer is computed (Line 17). Then, the backward pass computation (Line 18 to 27) proceeds as follows: First, we load the saved parameters and input activations of the current layer (Line 19). Next, we perform the backward pass to generate the input activation gradients and weight gradients using `backward_input` and `backward_weight` functions, which correspond to Equations 2 and 3 in the paper, respectively. As sparse input activations and parameters are saved in the forward pass the computation is sparse.

After the backward pass of the current mini-batch, the optimizer uses the computed weight gradients to update the parameters (Line 28 to 30).

## A.2 CIFAR-100

### A.2.1 Unstructured SWAT

Table 1 compares SWAT-U, SWAT-ERK and SWAT-M with unstructured sparsity for VGG-16, WRN-16-8 and DenseNet-121 architecture on CIFAR-100 dataset. The training procedure is the same as outlined in Section 4.1 in the paper. Hyperparameters are listed in Appendix subsection A.10.

Table 1: Unstructured SWAT on CIFAR-100 dataset.

| Network | Methods | Training Sparsity | | Top-1 | |
| --- | --- | --- | --- | --- | --- |
| | | Weight (%) | Activation (%) | Accuracy (%) | Accuracy Change |
| VGG-16 | SWAT-U | 90.0 | 90.0 | 69.8 | -2.3 |
| | SWAT-ERK | 90.0 | 69.6 | 71.8 | -0.3 |
| | SWAT-M | 90.0 | 59.9 | 72.2 | +0.1 |
| WRN-16-8 | SWAT-U | 90.0 | 90.0 | 77.6 | -1.7 |
| | SWAT-ERK | 90.0 | 77.6 | 78.5 | -0.8 |
| | SWAT-M | 90.0 | 73.3 | 77.9 | -1.4 |
| DENSENET-121 | SWAT-U | 90.0 | 90.0 | 77.2 | -0.4 |
| | SWAT-ERK | 90.0 | 90.0 | 76.5 | -1.1 |
| | SWAT-M | 90.0 | 84.2 | 75.5 | -2.1 |

### A.2.2 Structured SWAT

Table 2 compares SWAT-U with unstructured sparsity for ResNet-18 and DenseNet-121 architecture on CIFAR-100 dataset. The training procedure is the same as outlined in Section 4.1 in the paper. Hyperparameters are listed in Appendix subsection A.10.

Table 2: Structured SWAT on CIFAR-100 dataset.

| Network | Methods | Training Sparsity | | | Top-1 | |
|---------|---------|-------------------|----------------|-----------------------|-----------------|-----------------|
| | | Weight (%) | Activation (%) | Channel Pruned (%) | Accuracy (%) | Accuracy Change |
| RESNET-18 | SWAT-U | 50.0 | 50.0 | 50.0 | 76.4 | -0.4 |
| | | 60.0 | 60.0 | 60.0 | 76.2 | -0.6 |
| | | 70.0 | 70.0 | 70.0 | 75.6 | -1.2 |
| DENSENET-121 | SWAT-U | 50.0 | 50.0 | 50.0 | 78.7 | +0.9 |
| | | 60.0 | 60.0 | 60.0 | 78.5 | +0.4 |
| | | 70.0 | 70.0 | 70.0 | 78.1 | +0.3 |

## A.3 FLOP Calculation

Consider a convolution layer with input tensor $X \in \mathbb{R}^{N \times C \times X \times Y}$ and weight tensor $W \in \mathbb{R}^{F \times C \times R \times S}$ to produce output tensor $O \in \mathbb{R}^{N \times F \times H \times W}$. Input tensor has $N$ samples; each sample has $C$ input channels of dimension $X \times Y$. Weight tensor has $F$ filters and each filters has $C$ channels of dimension $R \times S$. Output tensor has $N$ output samples and each sample has $F$ output channels of dimension $H \times W$.

During the forward pass, input tensor is convolved with weight tensor to produce output tensor. In contrast, in the backward pass, the error-gradient of output tensor is deconvolved with input and weight tensor to produce weight gradient and input gradient respectively. The forward pass FLOP calculation assumes $s1$ sparsity in weight tensor. The effect of default sparsity in activation for forward pass computation is ignored since the default activation sparsity is present for both sparse learning and SWAT algorithms. However, for the backward pass FLOP calculation, since for SWAT the activation is explicitly sparsified therefore the FLOP calculation for SWAT is done using $s1$ weight sparsity and $s2$ activation sparsity whereas for sparse learning algorithms, $s1$ weight sparsity and default activation sparsity is assumed. The default activation sparsity generally vary between $30 - 50\%$, for our calculation we assumed default activation sparsity of 50%.

All the sparse learning algorithms and SWAT require some extra FLOP for connectivity update and regrowth connection such as dropping low magnitude component and thresholding. We omit the FLOP needed for these operations in our training FLOP calculation. For dynamic sparse learning algorithms such as SNFS [4], DSR [8] and DST [7], the weight sparsity varies during iterations and therefore we computed the average weight sparsity for different layers during the entire training and used it for computing the training FLOP.

### A.3.1 Computation in Convolution Layer

**Forward Pass**
Filter from the weight tensor is convolved with some sub-volume of input tensor of dimension $X^{sub} \in \mathbb{R}^{C \times R \times S}$ to produce a single value in output tensor. Therefore, the total FLOP for any single value in output tensor is $C \times R \times S$ floating-point multiplication $+ C \times R \times S - 1$ floating-point addition. This is approximately equal to $C \times R \times S$ floating-point MAC operations. Note, here 1 floating-point MAC operations = 1 floating-point multiplication + 1 floating-point addition. Thus, the total computation in the forward pass is equal to $(C \times R \times S) \times N \times F \times H \times W$ MAC operations.

Now, lets assume weight tensor is sparse. The overall sparsity in the weight tensor is $s1$ and the sparsity per filter in the weight tensor is $s^1, s^2.....,s^F$, for F filters in the layer. Note, $s1 = \frac{\sum_{x=1}^{F} s^x}{F}$. Theoretically only the non-zero weight will contribute to the FLOP. Therefore, the total FLOP for a single value in output tensor produce by the filter $x$ is $s^x \times C \times R \times S$ MAC operations. Thus, the total FLOP contribution for producing an output channel, of dimension $H \times W$, by the filter $x$ is $s^x \times (C \times R \times S) \times H \times W$. Therefore, the total FLOP for N input batches and F output channel is equal to $\sum_{x=1}^{F} (s^x \times (C \times R \times S) \times N \times H \times W) = (\sum_{x=1}^{F} s^x) \times (C \times R \times S) \times N \times H \times W = s1 \times F \times (C \times R \times S) \times N \times H \times W$. Hence, theoretically the FLOP reduction is proportional to sparsity in weight tensor.

$$\textbf{Forward Pass FLOP} = s1 \times F \times (C \times R \times S) \times N \times H \times W. \tag{1}$$

**Backward Pass**

For the backward pass, we back-propagate the error signal for computing the gradient of parameters. During the backward pass, each layer calculates 2 quantities: input gradient and weight gradient.

For the input gradient computation, the output gradient is deconvolved with filters to produce the input gradient. It can be implemented by rotating the filter tensor from $W \in \mathbb{R}^{F \times C \times R \times S}$ to $W \in \mathbb{R}^{C \times F \times R \times S}$ and convolving with output gradient. In this computation, the convolution data is output gradient and convolution kernels are filters. Therefore, as described in the previous section, the total MAC in the input gradient is approximately proportional to $(F \times R \times S) \times N \times C \times X \times Y$. Hence, the FLOP reduction with sparsity $s1$ in weight tensor will be approximately proportional to $s1 \times C \times (F \times R \times S) \times N \times X \times Y$.

$$\textbf{Input Gradient FLOP} = s1 \times C \times (F \times R \times S) \times N \times X \times Y. \tag{2}$$

For the weight gradient computation, the input activation is deconvolved with the output gradient to produce the weight gradient. It can be implemented by rotating the output gradient tensor from $\bigtriangledown O \in \mathbb{R}^{N \times F \times H \times W}$ to $\bigtriangledown O \in \mathbb{R}^{F \times N \times H \times W}$ and input actvation from $X \in \mathbb{R}^{N \times C \times X \times Y}$ $X \in \mathbb{R}^{C \times N \times X \times Y}$. The rotated output gradient is convolved on input activation to produce weight gradient. In this computation, the convolution data is input activation and convolution kernel is output gradient. Therefore, the total MAC in the weight gradient is approximately proportional to $(N \times X \times Y) \times F \times C \times R \times S$. Hence, as described in the previous section, the FLOP reduction with sparsity $s2$ in input activation tensor will be approximately proportional to $s2 \times (N \times X \times Y) \times F \times C \times R \times S$.

$$\textbf{Weight Gradient FLOP} = s2 \times (N \times X \times Y) \times F \times C \times R \times S. \tag{3}$$

Thus, the computational expense of the backward pass is approximately twice that of forward pass.

### A.3.2 Computation in Linear Layer

Consider a linear layer with input tensor $X \in \mathbb{R}^{N \times X}$ and weight tensor $W \in \mathbb{R}^{X \times Y}$ to produce output tensor $O \in \mathbb{R}^{N \times Y}$. During the forward pass, input tensor is multiplied with weight tensor to produce output tensor. In contrast, in the backward pass, the error-gradient of output tensor is multiplied with input and weight tensor to produce error-gradient of weight and error-gradient input tensor.

**Forward Pass**

The total FLOP for the forward pass is $N \times Y \times X$ floating-point multiplication + $N \times Y \times (X - 1)$ floating-point addition. This is approximately equal to $N \times X \times Y$ floating-point MAC operations.

Now, lets assume weight tensor is sparse. The overall sparsity in the weight tensor is $s$ and the sparsity per column in the weight tensor is $s^1, s^2.....s^Y$ for Y columns in the weight tensor. Note, $s = \frac{\sum_{y=1}^{Y} s^y}{Y}$. Theoretically only the non-zero weight will contribute to the FLOP. Therefore, the total FLOP for a single value in the $y$ column of output tensor is $s^y \times X$ MAC operations. Thus, the total FLOP for all the element in the $y$ column of output tensor is $s^y \times X \times N$. Therefore, the FLOP in forward pass is equal to $\sum_{y=1}^{Y} s^y \times N \times Y = N \times X \times (\sum_{x=1}^{F} s^x) = s \times N \times X \times Y$. Thus, theoretically the FLOP reduction will be proportional to sparsity in weight tensor.

**Backward Pass**

The computational expense of the backward pass is twice that of forward pass and is reduced proportionally to the sparsity of weight and activation tensor.

### A.4 Top-K Overhead

The Top-K operation can be efficiently implemented by finding the K-th largest element using an introselect algorithm and performing thresholding operation over the tensor. To estimate the overhead of finding the K-th largest element, we use numpy.partition function. The numpy.partition function uses the introselect algorithm and rearranges the array such that the element in the K-th position is in the position it would be in the sorted array. This overhead would be larger than the cost of finding the K-th largest element due to the extra rearrangement operations and thus serves as an upper bound on Top-K overhead.

We profile the numpy partition function using Vtune 2020.1.0.607630 on Intel(R) Core(TM) i9-7920X CPU @ 2.90GHz. Table 3 shows the total number of retired instruction while executing numpy.partition for different array size. We can observe that the overhead of the numpy.partition increases linearly with array size.

Table 3: Overhead of computing the K-th largest element using introselect algorithm

| Array Size | Retired Instruction | Retired Instruction/Size |
|---|---|---|
| 1000 | 15215 | 15.2 |
| 10000 | 217863 | 21.8 |
| 100000 | 1381744 | 13.8 |
| 1000000 | 12454369 | 12.6 |

The Top-K function in Pytorch framework is not an optimized implementation for GPU. Therefore, to estimate the Top-K overhead on GPU, we used the efficient Radix based Top-K implementation by Shanbhag et al. [10]. Table 4 shows the overall overhead of performing the Top-K operation for ResNet-50 on the ImageNet dataset with a batch-size of 32 on NVIDIA RTX-2080 Ti.

Table 4: Top-K Overhead

| Top-K Weight | Top-K Activation | Forward + Backward Pass | Total Top-K Overhead |
|---|---|---|---|
| 18 ms | 452 ms | 147 ms | 470 ms |

SWAT-U reduces the training FLOP per iteration by 76.1% and 85.6% at 80% and 90% sparsity respectively. Therefore, theoretical training speed up (assuming hardware can directly translate FLOPs reduction into reduced execution time) for the SWAT-U algorithm with a Top-K period of 1000 iteration, at 80%, and 90% sparsity would be $4.13\times$ and $6.79\times$ respectively. These numbers are found as follows:

$$\text{Speed Up at 80\% sparsity} = \frac{1000 \times 147}{999 \times (1 - 0.761) \times 147 + 147 \times (1 - 0.761) + 470} = 4.13 \quad (4)$$

$$\text{Speed Up at 90\% sparsity} = \frac{1000 \times 147}{999 \times (1 - 0.856) \times 147 + 147 \times (1 - 0.856) + 470} = 6.79 \quad (5)$$

The are additional optimizations that could potentially be applied to further reduce the Top-K overhead which we have not yet evaluated: (1) The overhead reported in Table 4 is for a Top-K operation, whereas we only need to find a threshold, not the top-K weights or activation values and then we can use that threshold for many iterations as suggested by the data in Figure 4 in the paper. So a more efficient implementation should be possible using the K-Selection Algorithm for which efficient GPU implementations have been proposed [1]; (2) There are more efficient *approximate* Top-K algorithms [2]; (3) Given a slow rate of change in threshold values per iteration, we could potentially **hide** the latency of the Top-K or K-Selection operation during the longer Top-K sampling period by starting the operation earlier. I.e., we can potentially exploit the stability in thresholds to move obtaining computation of updates to thresholds (after the first iteration) from the critical path; (4) The overhead of finding the Top-K operation on activations in Table 4 is higher due to large activation size. We speculate this overhead could be reduced significantly by performing the Top-K operation on activation for a single sample and using the resulting threshold for computing the approximate Top-K operation for the entire batch.

## A.5 Top-K Selection

Given CNNs operate on tensors with many dimensions, there are several options for how to select which components are set to zero during sparsification. Our CNNs operate on fourth-order tensors, $T \in R^{N \times C \times H \times W}$. Below we evaluate three variants of the Top-K operation illustrated in the right side of Figure 1. We also compared against a null hypothesis in which randomly selected components of a tensor are set to zero.

Figure 1: **Different ways of performing top-k operation**. 'N' denotes the #samples in the mini-batch or filters in the layer, 'C' denotes the #channels in the layer. 'H' and 'W' denote the height and width of the filter/activation map in the layer. Color represent the selected activations/weights by the Top-K operation.

The first variant, labeled TOPK-NCHW in Figure 1, selects activations and weights to set to zero by considering the entire mini-batch. This variant performs Top-K operation over the entire tensor, $f_{TOPK}^{\{N,C,H,W\}}(T)$, where the superscript represents the dimension along which the Top-K operation is performed. The second variant (TOPK-CHW) performs Top-K operation over the dimensions $C$, $H$ and $W$ i.e., $f_{TOPK}^{\{C,H,W\}}(T)$ , i.e., selects K % of input activations from every mini-batch sample and K% of weights from every filter in the layer. The third variant (TOPK-HW) is the strictest form of Top-K operation. It select K% of activations or weights from all channels, and thereby performing the Top-K operation over the dimension $H$ and $W$, i.e., $f_{TOPK}^{\{H,W\}}(T_{H,W})$.

The left side of Figure 1 shows the accuracy achieved on ResNet-18 for CIFAR100 when using SAW (see Appendix A.7) configured with each of these Top-K variants along with a variant where a random subset of components is set to zero. The results show, first, that randomly selecting works only for low sparsity. At high sparsity all variants of Top-K outperform random selection by a considerable margin. Second, they show that the more constrained the Top-K operation the less accuracy achieved. Constraining Top-K results in selecting some activations or weights which are quite small. Similarly, some essential activations and weights are discarded just to satisfy the constraint.

## A.6    Periodic Top-K

We have shown there is a little variation in the 'K-th' largest element during training, and it remains approximately constant as training proceed. Therefore, the Top-K does not need to be computed every iteration and can be periodically computed after some iterations. We define the number of iterations between computing the threshold for Top-K as the "Top-K period". Since the periodic Top-K used the same threshold during the entire period, therefore, it is crucial to confirm that periodic Top-K implementation does not adversely affect the sparsity during training. We dumped the amount of sparsity obtained in weights and activation using periodic Top-K with period 100 iteration with target sparsity of 90%. Figure 2 shows the sparsity during training using periodic Top-K implementation is concentrated around our targeted sparsity, and the fluctuation decreases as training proceeds confirming our hypothesis that chosen Top-K parameter stabilizes i.e. the Top-K threshold converge to a fixed value during the latter epochs.

## A.7    Sparsification of Output Gradients During Back-Propagation

SWAT is different from meProp as it uses sparse weight and activation during back-propagation, whereas meProp uses sparse output gradients. Our sensitivity analysis shows that convergence is extremely sensitive to the sparsification of output gradients. We compare the performance of the meProp and SWAT with deep networks and complex datasets. To compare SWAT's approach to that of meProp, we use a variant of SWAT-U that only sparsifies the backward pass; we shall refer to this version of SWAT-U as SAW (Sparse Activation and Weight back-propagation). Figure 3 shows SAW and meProp convergence of ResNet18 with the ImageNet dataset. It compares the performance of meProp at 30% and 50% sparsity to SAW at 80% sparsity. As we can see, meProp converges to a good solution at sparsity of 30%. However, at 50% sparsity, meProp suffers from overfitting and fails to generalize (between epochs 5 to 30), and at the same time, it is unable to reach an accuracy level

Figure 2: Sparsity Variation using Periodic Top-K Implementation. Network: ResNet-18, Dataset: CIFAR100, Top-K period: 100 iterations, Target Sparsity: 90%

above 45%. These results suggest that dropping output activation gradient ($\bigtriangledown_{a_l}$) is generally harmful during back-propagation. On the other hand, SAW succeeds to converge to a higher accuracy even at a sparsity of 80%.

Moreover, SWAT uses sparse weights and activations in the backward pass allowing compression of weights and activations in the forward pass. Effectively, reducing overall memory access overhead of fetching weights in the backward pass and activations storage overhead because only Top-K% activations are saved. This memory benefit is not present for meProp since dense weights and activations are needed in the backward pass, whereas there is no storage benefit of sparsifying the output gradients since they are temporary value s generated during back-propagation.

### A.8 Sparsification of Batch-Normalization Layer:

The activations and weights of BN layers are not sparsified in SWAT. Empirically, we found that sparsifying weights and activations are harmful to convergence. This is because the weight (gamma) of BN layers is a scaling factor for an entire output channel, therefore, making even a single BN weight (gamma) zero makes the entire output channel zero. Similarly, dropping activations affects the mean and variance computed by BN. Empirically we found that the BN layer is extremely sensitive to changes in the per channel mean and variance. For example, when ResNet18 is trained on CIFAR 100 using SWAT with 70% sparsity and we sparsify the BN layer activations, accuracy is degraded by 4.9% compared to training with SWAT without sparsifying the BN layers. Therefore, the activations of batch-normalization layer are not sparsified.

The parameters in a BN layer constitute less than 1.01% to the total parameters in the network and the total computation in the BN layer is less than 0.8% of the total computation in one forward and backward pass. Therefore, not sparsifying batch-normalization layers only affects the activation overhead in the backward pass.

Figure 3: **Convergence Analysis**: Shows the training curve of ResNet18 on ImageNet for meProp and SAW algorithm. Learning rate is reduced by $\frac{1}{10}^{th}$ at $30^{th}$ and $40^{th}$ epoch.

### A.9 Workstation Description

| WORKSTATION-DESCRIPTION | |
|---|---|
| CPU | Intel(R) Core(TM) i9-9900X CPU @ 3.50GHz |
| | Intel(R) Xeon(R) Silver 4116 CPU @ 2.10GHz |
| GPU | NVIDIA 2080-Ti |
| UBUNTU | Ubuntu 18.04.2 LTS |
| NVIDIA-DRIVER | 440.33.01 , 418.43 |
| CUDA, cuDNN | CUDA==10.0.130, cuDNN==7.501 |
| Pytorch | pytorch==1.1.0, torchvision==0.3.0 |

### A.10 Details of implementation

We implemented all models and algorithms on `pytorch` framework. Code can be found at `https://github.com/AamirRaihan/SWAT`. To ease the reproducibility of our experiments, we have also created a docker image. We have also uploaded the model checkpoint on anonymous dropbox folder for easily verifying the trained model `https://www.dropbox.com/sh/vo4dxuogk4Οn6mg/AACdCWWhkhsYdqpjuvsvIb5Οa?dl=0`.

Table 5: Hyperparameters for ResNet, VGG and DenseNet experiments on CIFAR10/100

| Experiment | ResNet 18, 50, 101 | | VGG 16 | | DenseNet BC-121 | |
|---|---|---|---|---|---|---|
| Number of training epochs | | 150 | | 150 | | 150 |
| Mini-batch size (#GPU) | | 128 (1) | | 128 (1) | | 64 (1) |
| Learning rate schedule (epoch range: learning rate) | 1 - 50: 51 - 100: 101- 150 : | 0.100 0.010 0.001 | 1 - 50: 51 - 100: 101- 150 : | 0.100 0.010 0.001 | 1 - 50: 51 - 100: 101- 150 : | 0.100 0.010 0.001 |
| Optimizer | SGD with Momentum | | SGD with Momentum | | SGD with Momentum | |
| Momentum | | 0.9 | | 0.9 | | 0.9 |
| Nesterov Acceleration | | False | | False | | False |
| Weight Decay | | 5e-4 | | 5e-4 | | 5e-4 |
| TopK Implementation | | TopK-NCHW | | TopK-NCHW | | TopK-NCHW |
| TopK Period | | Per Iteration | | Per Iteration | | Per Iteration |

Table 6: Hyperparameters for WideResNet experiments on CIFAR10/100

| Experiment | WideResNet Depth=28 Widen Factor=10 | |
|---|---|---|
| Number of training epochs | | 200 |
| Mini-batch size (#GPU) | | 128 (1) |
| Learning rate schedule (epoch range: learning rate) | 1 - 60: 61 - 120: 121- 160 : 161- 200 : | 0.100 0.020 0.004 0.0008 |
| Optimizer | | SGD with Momentum |
| Momentum | | 0.9 |
| Nesterov Acceleration | | True |
| Weight Decay | | 5e-4 |
| Dropout Rate | | 0.3 |
| TopK Implementation | | TopK-NCHW |
| TopK Period | | Per Iteration |
| Remarks | First and Last layer are not sparsified since the total parameters in these layers are less than 0.17% of the total network parameters | |

Table 7: Hyperparameters for ResNet50/WRN-50-2 experiments on ImageNet

| Experiment | SWAT(UnStructured) | | SWAT(Structured) | |
|---|---|---|---|---|
| Number of training epochs | | 90 | | 90 |
| Mini-batch size (#GPU) | | 256 (8) | | 256 (8) |
| Learning rate schedule (epoch range: learning rate) | 1 - 30:<br>31 - 60:<br>61- 80 :<br>81- 90 : | 0.100<br>0.010<br>0.001<br>0.0001 | 1 - 30:<br>31 - 60:<br>61- 80 :<br>81- 90 : | 0.100<br>0.010<br>0.001<br>0.0001 |
| Learning Rate WarmUp | | Linear (5 Epochs) | | Linear (5 Epochs) |
| Optimizer | | SGD with Momentum | | SGD with Momentum |
| Momentum | | 0.9 | | 0.9 |
| Nesterov Acceleration | | True | | True |
| Weight Decay | | 1e-4 | | 1e-4 |
| Weight Decay on BN parameters | | No | | No |
| Label Smoothing | | 0.1 | | 0.1 |
| TopK Implementation | | TopK-NCHW | | TopK-Channel |
| TopK-Period | | 1000 iteration | | 1000 iteration |
| Remarks | First is not sparsified due to low parameters count. To speed up training, we used efficient Top-K implementation where the Top-K is computed periodically after 1000 iteration | | | |

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

[Supplementary Material 2 · README.pdf]

# DRAMSim2

Elliott Cooper-Balis
Paul Rosenfeld
Bruce Jacob
University of Maryland
dramninjas *[at]* gmail *[dot]* com

## Contents

## 1 About DRAMSim2

DRAMSim2 is a cycle accurate model of a DRAM memory controller, the DRAM modules which comprise system storage, and the buses by which they communicate.

The overarching goal is to have a simulator that is small, portable, and accurate. The simulator core has a simple interface which allows it to be CPU simulator agnostic and should to work with any simulator (see section 4.2). This core has no external run time or build time dependencies and has been tested with g++ on Linux as well as g++ on Cygwin on Windows.

## 2    Getting DRAMSim2

DRAMSim2 is available on github. If you have git installed you can clone our repository by typing:

```
$ git clone git://github.com/dramninjasUMD/DRAMSim2.git
```

## 3    Building DRAMSim2

To build an optimized standalone trace-based simulator called `DRAMSim` simply type:

```
$ make
```

For a debug build which contains debugging symbols and verbose output, run:

```
$ make DEBUG=1
```

To build the DRAMSim2 library, type:

```
$ make libdramsim.so
```

# 4 Running DRAMSim2

## 4.1 Trace-Based Simulation

In standalone mode, DRAMSim2 can simulate memory system traces. While traces are not as accurate as a real CPU model driving the memory model, they are convenient since they can be generated in a number of different ways (instrumentation, hardware traces, CPU simulation, etc.) and reused.

We've provided a few small sample traces in the `traces/` directory. These gzipped traces should first be preprocessed before running through the simulator. To run the preprocessor (the preprocessor requires python):

```
cd traces/
./traceParse.py k6_aoe_02_short.trc.gz
```

This should produce the file `traces/k6_aoe_02_short.trc`. Then, go back to the DRAMSim2 directory and run the trace based simulator:

```
cd .
./DRAMSim -t traces/k6_aoe_02_short.trc -s system.ini -d ini/DDR3_micron_64M_8B_x4_sg15.ini -c 1000
```

This will run a 1000 cycle simulation of the `k6_aoe_02_short` trace using the specified DDR3 part. The -s, -d, and -t flags are required to run a simulation.

A full list of the command line arguments can be obtained by typing:

```
$ ./DRAMSim --help
DRAMSim2 Usage:
DRAMSim -t tracefile -s system.ini -d ini/device.ini [-c #] [-p pwd] -q
  -t, --tracefile=FILENAME  specify a tracefile to run
  -s, --systemini=FILENAME  specify an ini file that describes the memory system parameters
  -d, --deviceini=FILENAME  specify an ini file that describes the device-level parameters
  -c, --numcycles=#      specify number of cycles to run the simulation for [default=30]
  -q, --quiet         flag to suppress simulation output (except final stats) [default=no]
  -o, --option=OPTION_A=234     overwrite any ini file option from the command line
  -p, --pwd=DIRECTORY   Set the working directory
```

Some traces include timing information, which can be used by the simulator or ignored. The benefit of ignoring timing information is that requests will stream as fast as possible into the memory system and can serve as a good stress test. To toggle the use of clock cycles, please change the `useClockCycle` flag in `TraceBasedSim.cpp`. If you have a custom trace format you'd like to use, you can modify the `parseTraceFileLine()` function ton add support for your own trace formats.

The prefix of the filename determines which type of trace this function will use (ex: k6_foo.trc) will use the k6 format in `parseTraceFileLine()`.

## 4.2 Library Interface

In addition to simulating memory traces, DRAMSim2 can also be built as a dynamic shared library which is convenient for connecting it to CPU simulators or other custom front ends. A `MemorySystem` object encapsulates the functionality of the memory system (i.e., the memory controller and DIMMs). The classes that comprise DRAMSim2 can be seen in figure 1. A simple example application is provided in the `example_app/` directory. At this time we have plans to provide code to integrate DRAMSim2 into MARSSx86, SST, and (eventually) M5.

Figure 1: Block diagram of DRAMSim2. The `recv()` functions are actually called `receiveFromBus()` but were abbreviated to save sapce.

# 5 Example Output

The verbosity of the DRAMSim2 can be customized in the system.ini file by turning the various debug flags on or off.

Below, we have provided a detailed explanation of the simulator output. With all DEBUG flags enabled, the following output is displayed for each cycle executed.

**NOTE** : BP = Bus Packet, T = Transaction

MC = MemoryController, R# = Rank (index #)

```
----------------- Memory System Update -----------------
---------- Memory Controller Update Starting ------------ [8]
 -- R0 Receiving On Bus    : BP [ACT] pa[0x5dec7f0] r[0] b[3] row[1502] col[799]
 -- MC Issuing On Data Bus    : BP [DATA] pa[0x7edc7e0] r[0] b[2] row[2029] col[799] data[0]=
 ++ Adding Read energy to total energy
 -- MC Issuing On Command Bus : BP [READ_P] pa[0x5dec7f8] r[1] b[3] row[1502] col[799]
== New Transaction - Mapping Address [0x5dec800] (read)
  Rank : 0
  Bank : 0
  Row  : 1502
  Col  : 800
 ++ Adding IDD3N to total energy [from rank 0]
 ++ Adding IDD3N to total energy [from rank 1]
== Printing transaction queue
  8]T [Read] [0x45bbfa4]
  9]T [Write] [0x55fbfa0] [5439E]
  10]T [Write] [0x55fbfa8] [1111]
== Printing bank states (According to MC)
[idle] [idle] [2029] [1502]
[idle] [idle] [1502] [1502]

== Printing Per Rank, Per Bank Queue
 = Rank 0
    Bank 0   size : 2
       0]BP [ACT] pa[0x5dec800] r[0] b[0] row[1502] col[800]
       1]BP [READ_P] pa[0x5dec800] r[0] b[0] row[1502] col[800]
    Bank 1   size : 2
       0]BP [ACT] pa[0x5dec810] r[0] b[1] row[1502] col[800]
       1]BP [READ_P] pa[0x5dec810] r[0] b[1] row[1502] col[800]
    Bank 2   size : 2
       0]BP [ACT] pa[0x5dec7e0] r[0] b[2] row[1502] col[799]
       1]BP [READ_P] pa[0x5dec7e0] r[0] b[2] row[1502] col[799]
    Bank 3   size : 1
       0]BP [READ_P] pa[0x5dec7f0] r[0] b[3] row[1502] col[799]
 = Rank 1
    Bank 0   size : 2
       0]BP [ACT] pa[0x5dec808] r[1] b[0] row[1502] col[800]
       1]BP [READ_P] pa[0x5dec808] r[1] b[0] row[1502] col[800]
    Bank 1   size : 2
       0]BP [ACT] pa[0x5dec818] r[1] b[1] row[1502] col[800]
       1]BP [READ_P] pa[0x5dec818] r[1] b[1] row[1502] col[800]
    Bank 2   size : 1
       0]BP [READ_P] pa[0x5dec7e8] r[1] b[2] row[1502] col[799]
    Bank 3   size : 0
```

Anything sent on the bus is encapsulated in an BusPacket (BP) object. When printing, they display the following information:

```
BP [ACT] pa[0x5dec818] r[1] b[1] row[1502] col[800]
```

The information displayed is (in order): command type, physical address, rank #, bank #, row #, and column #.

Lines beginning with " − " indicate bus traffic, ie,

```
-- R0 Receiving On Bus       : BP [ACT] pa[0x5dec7f0] r[0] b[3] row[1502] col[799]
-- MC Issuing On Data Bus    : BP [DATA] pa[0x7edc7e0] r[0] b[2] row[2029] col[799] data[0]=
-- MC Issuing On Command Bus : BP [READ_P] pa[0x5dec7f8] r[1] b[3] row[1502] col[799]
```

Sender and receiver are indicated and the packet being sent is detailed.

Lines beginning with " ++ " indicate power calculations, ie,

```
++ Adding Read energy to total energy
++ Adding IDD3N to total energy [from rank 0]
++ Adding IDD3N to total energy [from rank 1]
```

The state of the system and the actions taken determine which current draw is used. For further detail about each current value, see Micron datasheet.

If a pending transaction is in the transaction queue, it will be printed, as seen below:

```
== Printing transaction queue
1]T [Read] [0x45bbfa4]
2]T [Write] [0x55fbfa0] [5439E]
3]T [Write] [0x55fbfa8] [1111]
```

Currently, at the start of every cycle, the head of the transaction queue is removed, broken up into DRAM commands and placed in the appropriate command queues. To do this, an address mapping scheme is applied to the transaction's physical address, the output of which is seen below:

```
== New Transaction - Mapping Address [0x5dec800] (read)
 Rank : 0
   Bank : 0
   Row  : 1502
   Col  : 800
```

If there are pending commands in the command queue, they will be printed. The output is dependent on the designated structure for the command queue. For example, per-rank/per-bank queues are shown below:

```
= Rank 1
 Bank 0   size : 2
    0]BP [ACT] pa[0x5dec808] r[1] b[0] row[1502] col[800]
    1]BP [READ_P] pa[0x5dec808] r[1] b[0] row[1502] col[800]
 Bank 1   size : 2
    0]BP [ACT] pa[0x5dec818] r[1] b[1] row[1502] col[800]
    1]BP [READ_P] pa[0x5dec818] r[1] b[1] row[1502] col[800]
 Bank 2   size : 1
    0]BP [READ_P] pa[0x5dec7e8] r[1] b[2] row[1502] col[799]
 Bank 3   size : 0
```

The state of each bank in the system is also displayed:

```
== Printing bank states (According to MC)
[idle] [idle] [2029] [1502]
[idle] [idle] [1502] [1502]
```

Banks can be in many states, including idle, row active (shown with the row that is active), refreshing, or precharging. These states will update based on the commands being sent by the controller.

# 6   Results Output

In addition to printing memory statistics and debug information to standard out, DRAMSim2 also produces a 'vis' file in the results/ directory. A vis file is essentially a summary of relevant statistics that is generated per

epoch (the number of cycles per epoch can be set by changing the EPOCH_COUNT parameter in the system.ini file).

We are currently working on DRAMVis, which is a cross-platform viewer which parses the vis file and generates graphs that can be used to analyze and compare results.