[Reviews · NeurIPS 2020]

Review 1

Summary and Contributions: This paper proposed an efficient training framework. By experiments on different models, the paper showed that sparse weight in forward and sparse weight and activation in back propagation is the best for the accuracy-efficiency trade-off. And a magnitude-based pruning method is embedded in the framework to build unstructural or structural sparsity.

Strengths: 1. The experiments result of choices on which part to be sparse (Figure 1 and 2) is influential in the efficient training filed to help other researchers 2. The experiments are conducted in various models in CIFAR-10 and ImageNet, which demonstrate a stable improvement in the efficiency compared with the vanilla training.

Weaknesses: 1. Lack of Novelty: The idea of sparsifying weights and activations during training and the magnitude-based pruning is not novel. 2. Lack of sufficient explanation and analysis on the choice of which to be sparse: The experiments of setting which part in the forward and backward to be sparse (Figure 1 and 2) lack of sufficient explanation, which makes this paper have less insight for other researchers. 3. Not sufficient baselines are included in the experiments: The paper gave a rough comparison with previous work in Table 1, and not all previous works listed in Table 1 are included are the baselines in the experiments and the author didn’t give explanation on that. (e.g., DSG [29] is the most similar with the proposed SWAT based on Table 1 but it didn’t appear in the experiments as a baseline)

Correctness: Yes

Clarity: Yes

Relation to Prior Work: Yes

Reproducibility: Yes

Additional Feedback: More details on the Weaknesses ===After Rebuttal=== I read the authors' rebuttal and value their work in preparing it. However, major concerns remain.


Review 2

Summary and Contributions: This paper proposes a relatively novel CNN training routine called Sparse Weight Activation Training (SWAT). SWAT is aimed to be more compute and memory-efficient than the conventional CNN training while also inducing sparsity for the targetted model compression. SWAT's efficiency in training comes from two observations 1) that pruning of low-magnitude weights during the forward pass (not a new thing) does affect accuracy for the aimed sparsity and helps in compute reduction 2) removal of low-magnitude weights and activations during backprop thus leading to sparse mat-vec ops ensuring compute and memory efficiency. This is in contrast to making the gradients sparse by pruning the gradients as they observed that hurts accuracy. The second point is the new thing, even though not completely. The paper has good experimental evaluation on a variety of datasets including CIFAR-10/100, ImageNet using multiple standard architectures like ResNets, DenseNets, VGG, and WideResNet. The paper also benchmarks the training speed up on a simulated sparse learning accelerator to showcase the gains with some loss in model accuracy. Lastly, SWAT also results in a memory-efficient backprop because of insight 2). In my opinion, while the end-to-end results are sound and good, most of the ideas do exist and sometimes even utilized to an extent in practice. I will expand on this in the next sections, but I think there is a lot going on in this paper with a very fundamental idea.

Strengths: The paper is very well motivated and compute and memory-efficient training is a very important thing to worry about. Most people have focused on the compute and memory efficiency of the inference. I have to point out that some of those methods indirectly or directly also tackled the efficient training aspect as briefly noted even in this paper. I will go sequentially The authors openly admit sparse training is still budding which is appreciated rather than making superfluous claims. The authors did a good job in the related work barring some minor omissions which I will discuss in the next section. Table 1 is a good summarization. I really like the simple experiments in Figure 1 and Figure 2 to showcase why SWAT needs to do what it is doing. This gives a good argument against using a sparse output gradient in the backward pass. However, I am a bit skeptical about the drastic drop due to sparse activations in forward pass when filter pruning methods do it well and same concerns for the sparse gradients, will talk in the weakness section. The fundamental idea of the sparse forward pass and use of the same sparse weights along with sparse activations is a good idea when the sparsities are high enough to get the gains. Good to see non-uniform budgets being used as well. I would also like to point out to STR (Kusupati et al., ICML 2020) which has sparsity budgets with lesser inference FLOPs so that SWAT could even improve the gains. This would be interesting to see. A good study in Figure 5 for the Top-K period sampling. The experiments on al the datasets are good albeit with some issues which I will note down in the next section. Even though not totally real-world, Table 6 with study in the simulator is a nice start. The study in Figure 7 is good evidence for the choice of dense gradients even though they could increase the compute cost. However, I have some concerns here as well and will talk in the next section. Figure 8 is valid and makes a lot of sense which is also observed in a lot of 3-phase pruning methods. The authors did a good job on the broader impact section and the plots' aesthetics.

Weaknesses: As I mentioned earlier, while the paper did a good job in doing what it did, all the ideas and sometimes even the combination of them existed before and have been shown effective. I will go sequentially again: In related work, the paper claims SWAT and Cai et al., suggested the use of L1 norm for channel/filter pruning, however, this idea already exists and is proven to be very effective for structured sparsity or group sparsity. An example paper would be Li et al., ICLR 2017 (https://arxiv.org/abs/1608.08710) there are more papers that followed up on it. Even though this talks about filters, extending it to channels of the output feature maps or the channels in each of the kernels is not a nontrivial thing). This is not a weakness, but there is a line of work on learnable sparsity which is similar to DST and more effective than it and can be found in Savarese et al., 2019 or Kusupati et al., 2020. It is good to see methods like RigL which rely on sparse-to-sparse training which also is a line of work which improve training efficiency along with a reduction in inference cost In table 1, the authors say network pruning methods depend on algorithms, but from my experience, most unstructured pruning methods can be adapted to structured pruning. This is just a comment. In Figure 1, I don't understand why sparse activations which is equivalent to filter-pruning (even though in a data-driven fashion) or even something very similar to targetted dropout performs so badly compared to unstructured pruning. I know this is trying for high levels of sparsity but for the networks being used for CIFAR-100, 90% sparsity doesn't even come close to that range where things start to drop off. I am not completely sure about this experiment. Figure 2 shows a very good point, but in my experience, I never faced such drastic drops from sparse output gradients. For example. GMP (Zhu & Gupta 2017), STR and CS (Savarese et al., 2019) have sparse gradients, do even RigL (Evci et al., 2020) apart from the occasional dense gradient. Even old schools hard threshold-based methods used for training sparse models like in Bonsai (Kumar et al., 2017) and FastGRNN (Kusupati et al., 2018) have sparse gradients with occasional dense gradients. Except for GMP, STR and CS, all the other methods are not gradual but have sparsity budget met when they get into that phase. I am not sure what is happening here. There is a method called DNW (Wortsman et al., NeurIPS 2019) which uses a sparse network for forward pass and dense gradients using a straight-through estimator in the backward pass and DNW does only as good as RigL with enough tuning. So dense vs sparse gradients are not well experimented and the same goes with Figure 7. I don't understand section 3.3.1. What is channel selection? Are you trying to remove different channels across each kernel/filter in the layer (I get the feeling that it is this from the picture). if it is the case, the gains would be much less than the former in accelerators. A typical channel pruning is applied such that same channel is removed from all the filters resulting in the input feature map channel reduction. I would like a clarification on this, maybe I am missing something. Yes, channel pruning in the way shown in the figure can still help for speedup, but needs to be well handled and is not as simple as the typical case. What is the base model accuracy in the experiment tables? I might have missed this, is there a dense phase for your method as well (I assume not). I am assuming default sparsity to be dense whenever DS is mentioned. Table 2: I am super surprised that ERK budget has lesser inference FLOPs than Uniform across all the methods. In my experience, ERK is expensive in terms of inference FLOPs even when the first layer of uniform is dense. A similar argument for training FLOPs. Also, how did you choose the activation sparsity in your experiments? Is there something being tuned there as it is not directly obtained from the sparsity budget. Table 3: You mentioned all the methods you compare against having dense trained models, but that is not true for methods apart from DST. How is DST having much lower training FLOPs compared to RigL when one is sparse-sparse and other is dense to sparse. The inference flops of RigL will be the same as SWAT-U if the budgets are uniform with first layer dense. Figure 6 caption is not at all clear and the x-axis just confuses the reader. My major problem is probably with Structured pruning experiments in Table 5. Most structured pruning methods (not the ones compared with offline pruning) involving channel/filter pruning all have sparse weights as the filters are pruned out and leading to sparse activations. If done right the structured sparsity methods do improve on training speeds as well. I request some clarification on this from the authors. This is not a weakness, as mentioned earlier, methods like STR came with better budgets than a uniform at similar accuracy, it would be good to see this speed up even more, this is just a suggestion for strengthening the paper.

Correctness: The proposed method is correct, however, I am not fully convinced with some experiments presented to argue the design choices. The empirical methodology is OK and can be further improved by focussing on right things rather than doing everything in a hodge-podge fashion.

Clarity: The paper is decently written, but at times becomes very dense or hard to follow. The paper can use another pass for both grammar and readability. The captions of certain figures should be more informative.

Relation to Prior Work: In related work, the paper claims SWAT and Cai et al., suggested the use of L1 norm for channel/filter pruning, however, this idea already exists and is proven to be very effective for structured sparsity or group sparsity. An example paper would be Li et al., ICLR 2017 (https://arxiv.org/abs/1608.08710) there are more papers that followed up on it. This is not a weakness, but there is a line of work on learnable sparsity which is similar to DST and more effective than it and can be found in Savarese et al., 2019 or Kusupati et al., 2020. It is good to see methods like RigL which rely on sparse-to-sparse training which also is a line of work which improve training efficiency along with a reduction in inference cost. I really liked the contrast with meProp and DSG. Related work even though extensive, is not complete due to the exploding field.

Reproducibility: Yes

Additional Feedback: I am very much willing to increase the score if the authors answer my key concerns on structured sparsity, the sparse gradients and sparse activation experiments along with other things mentioned above. This idea, due to the implementation, deserves to be published but is bogged down by other issues which are not well tackled. ------------------------------------------------------------------------------------------------------ ------------------------------------------------------------------------------------------------------ After Rebuttal: This was a comprehensive rebuttal. Thanks for the efforts. (1) The statement you made in the rebuttal is not clear in the paper. This means you are just temporarily masking the activations. Which is equivalent to Straight-through estimator. STE still performs competitively with optimizing network with sparse activations. In my experience, even this shouldn't result in such a drop if done right. But the explanation helps. Thanks. (2) This was a very useful clarification. So You look at the gradient and then sparsify it based on magnitudes. This probably makes sense, essentially saying the activations/filters connected to it which has little gradients might be getting no gradients to update. This overtime can mean that right filters don't update leading to drop in accuracy. Sorry for the misunderstanding, but this needs to be clarified in the paper. (3) Thanks (4) This is a bit problematic to me, selecting different channels for different filters means it is a as bad as unstrcutured sparsity and the locality and gains from just dropping filters/channels all together is gone. If we remove channel 3 from all of the filters then we can drop filter 3 from the earlier layer. It just helps in real deployment rather than then choice you made. (5) Thanks. This should be in the paper. Having basemodels on the top helps. (6) This is nice, I forgot about CIFAR. This makes sense. I understand DST helps, but DST is still dense to sparse so I am still confused how it beat RigL or SNFS in training FLOPs, unless it reduced inference flops by a lot, which it did not compared to them. (7) Thanks. Overall, I agree with other reviewers that it is a combination of known techniques. However, I like the insights and experimentation. I am increasing my score to marginally above acceptance because of the issues mentioned above. The rebuttal helped clarify some.


Review 3

Summary and Contributions: The authors propose a sparse training technique named Sparse Weight Activation Training (SWAT) in which both activations and weights are sparsified. On the forward pass, SWAT uses sparse weights and full activations. On the backwards pass, SWAT uses sparse weights and full gradients for the gradient propagation and sparse activations and full gradients for the weight update. All sparsity is induced by picking the top-K values, by layer. The authors test different variants of SWAT, which differ by how they pick the top-K threshold in each layer.

Strengths: Simple and realizable idea, clear and concise writing, large amount of experimental data. Figures 1 and 2 are very insightful. Huge amount of detail on FLOP computation, cycle count, and top-K practicality in the supplementary materials. Major effort in open sourcing code to enable reproducibility.

Weaknesses: There isn't much insight as to what specifically SWAP does better than the many related works. The list of technical contributions doesn't make it clear what the novelty is compared to related works. I don't get much from the experiment tables other than that SWAT outperforms the competition. To me, what makes the paper stand out is the effort put into validating the hardware performance of SWAT in the supplementary. I'd like to see some of this moved to the main paper. The gap between SWAP and the related work is small. On ImageNet the accuracy gap is less than 1%, and the FLOPs gap is about 9% (FLOPs don't automatically translate to hardware performance). EDIT: The rebuttal didn't change my mind about the lack of novelty. Regarding the performance gap, the authors discussed *theoretical* speedup which wasn't what I was talking about. I think my comments stand.

Correctness: Methodology for the the cycle count estimation is missing detail, even considering the supplementary. Does the simulator count memory cycles? Please add some of the text from the supplementary into the main paper, and perhaps add more detail of what's being simulated. Table 1: I have difficulties understanding some entries. - Based on Section 3.3, SWAT uses "full gradients" for both gradient propagation and weight updates. But the table says it uses sparse input gradients. - Based on Section 3.3, SWAT generates "dense weight gradients" but the table says it uses sparse weight gradients. - The table claims that only SWAT can be used for structured sparse pruning. It is difficult for me to believe that none of the related work can be adapted to prune channels instead of individual weights using L1 norm. This feels like a misleading claim. EDIT: The authors clarified Table 1, but I still feel that they should make a few changes to make things more clear. The authors provided a few details about the cycle count, but not enough to really convince me that it's trustworthy. Nevertheless there is enough compelling content in the paper and my score remains the same.

Clarity: Yes

Relation to Prior Work: Yes, though the novelty is a bit unclear to me from the list of technical contributions. There are a number of sparse training papers and it's not clear how SWAT stands out.

Reproducibility: Yes

Additional Feedback:


Review 4

Summary and Contributions: This paper introduces sparse weight activation training (SWAT), a method that sparsifies activations and weights to reduce the FLOPs of training. SWAT uses simple magnitude-based top-k selection for fine-grained pruning of weights and activations. The experiments show promising results in training acceleration in custom accelerators.

Strengths: Describe the strengths of the work. Typical criteria include: soundness of the claims (theoretical grounding, empirical evaluation), significance and novelty of the contribution, and relevance to the NeurIPS community.

Weaknesses: Although speeding up training by sparsifying both activations and weights may have previously lacked exploration, SWAT does not have substantial novelty. It simply applies magnitude-based top-k selection to both weights and activations. The authors suggested 3 distinct approaches to determine the sparsity ratios for each layer, which are not novel as they were introduced earlier by other publications. This paper is also missing important insights into how the different choices of configurations (fine-grained vs. corse-grained, sparsity ratios) impact training performance. SWAT is a hodgepodge of preexisting methods, without considering how they interact with each other. The authors missed the opportunity to explore the joint optimization of weight and activation sparsification.

Correctness: The method and empirical results appear to be correct. The paper contains the necessary details of the method and hyperparameters for reproducing the results. Part of the experiments were repeated to produce standard deviation.

Clarity: The paper is overall easy to follow with minor grammatical errors.

Relation to Prior Work: The authors differentiated their work from previous contributions by arguing that SWAT simultaneously sparsifies weights and activations. It concerns the reviewer that there might be previous attempts at this and SWAT might not be novel. Moreover, a simple combination of preexisting methods is not novel.

Reproducibility: Yes

Additional Feedback: It would substantially improve the contribution of this paper by proposing a new method that considers the opportunity of joint-optimization of weights and activations.

[Author Response · NeurIPS 2020]

We thank the reviewers for their feedback. Our paper will be updated to reflect the responses below.

**ALL:** *Novelty?* Our contribution are: (1) Demonstrating training is highly sensitive to sparsifying back-propagated output error gradients; (2) A simple training algorithm, sparse in forward and backward passes, with activation and weight sparsity that fits emerging sparse accelerators. *Why SWAT outperforms related work?* We believe this is because SWAT can perform dynamic topology exploration: Backpropagation with sparse weights and activations approximates backpropagation on a network with sparse connectivity and sparsely activated neurons. The gradients generated during back-propagation minimize loss for the current sparse connectivity. However, each iteration SWAT generates a potentially new sparse network using the sparsifying function. Non-active weights are also updated thus capturing fine-grained temporal importance of connectivity during training. Figure 7 and 8 show the importance of unmasked gradient updates and dynamic exploration of connectivity.

**Reviewer 1: (1)** When sparsifying Equation 2 and 3, we expect combinations sparsifying both gradients and weights and/or activations (i.e., (g,a), (g,w), (g,w,a)) would underperform as sparsifying gradients alone harms convergence. Also, as sparsifying weights or activations alone will not sparsify both equations, we skip these combinations too. **(2)** DSG loses considerable accuracy even at low sparsity. E.g., for ResNet18 on ImageNet at 50% sparsity DSG suffers an accuracy loss of 4.6%.

**Reviewer 2: (1)** *"Drastic drop due to sparse activations in forward pass"*: In Figure 1 we isolate the forward and backward pass and examine sensitivity of training to sparsifying only the forward pass. Notably, this means we use the full activation for the backward pass. So, in Figure 1 the backward pass is not optimizing the network for sparse activation. Typical filter pruning methods use sparse activation in the backward pass so the network adapts for activation sparsity. Kurtz et al. (ICML 2020) and Georgiadis (CVPR 2019) show how to increase activation sparsity in the forward pass. Combining such techniques with SWAT may reduce FLOPS without increasing error and thus may be a good direction for future work. **(2)** *"Never faced such drastic drops from sparse output gradients"*: We believe there is a misunderstanding: GMP, STR, CS and RigL use a sparse weight gradient during the parameter update stage whereas in Figure 2 we sparsify the back-propagated error gradient. The back-propagated output error gradient is sparsified before performing the convolutions to generate weight and input gradients. The generated weight gradient is dense because the convolution between a sparse activation and dense back-propagated error gradient tensor yields a dense result. This back-propagated error gradient is not dropped in STR, CS, and GMP but rather dense back-propagated error gradients are used to generate weight gradients that are masked during the parameter update stage. Thus, STR, CS, GMP only update the active parameters. **(3)** Yes, sparsity budget proposed by Kusupati et al. (ICML 2020) could be used for more speedup. **(4)** Yes, the different channels will be selected for different filters and there is overhead for introducing structured sparsity since the L1 response of channels is computed. However, the FLOP reduction due to high sparsity is more than the overhead of computing the L1 response. Moreover, the idea of periodically computing the Top-K channel selection is applicable. **(5)** SWAT doesn't have dense phase. Base model accuracy can be calculated from the accuracy drop. E.g., for ResNet-50 on ImageNet, the Top-1 accuracy is 76.8. **(6)** Yes, in general uniform will have lower flop than ERK especially when the input resolution is high. ERK generally applies lower sparsity at initial layer which have significant computation especially true for imagenet. However, in Table 2, ERK is more efficient because the initial layer has small computation due to small input resolution (CIFAR-10) and computationally expensive layers have higher sparsity. Similarly, we observe DST allocates higher sparsity to more computationally expensive layers and quickly reaches a sparsity where overall computation is low. **(7)** We will add discussion on Li et al. 2017 and follow-up work on structured sparsity.

**Reviewer 3: (1)** *The speedup gap is small:* The theoretical speedup by the earlier method would be around $\frac{1}{1-0.67} = 3.03\times$ whereas with our method the theoretical speedup would be $\frac{1}{1-0.76} = 4.16\times$. Theoretical speedup does not fully capture the benefit of SWAT since SWAT also reduces memory footprint during training. **(2)** *Methodology for cycle count estimation?* We will add relevant detail from the supplementary material. Simulator counts the cycles taken to spatially map and schedule the computation present in each layer. The memory hierarchy is similar to the DaDianNo architecture. *Table 1*: (i) The column represents whether the input gradient and weight gradient computation is sparse or not. (ii) Second, convolution between sparse input activation and dense back-propagated error gradient tensor generates a dense gradient for weight and the dense gradient is used in parameter update. (iii) Yes, related work can be adapted for structured sparsity.

**Reviewer 4**: Selecting the sparsity budget of SWAT (e.g., using ERK) is not our main contribution, but rather demonstrates SWAT can readily leverage such techniques. We have covered a range of configurations in Table 4, 5 6 and 7 in the main paper and Table 1 and 2 of the supplemental. Joint optimization of backward pass activation and weight sparsity is an good direction for future work.



[Meta-Review · NeurIPS 2020]

This paper proposes a simple heuristic for reducing the work and memory required for training networks. The heuristic is to TopK the weights and activations. Applying TopK to the weights isn't by itself novel, but the combination with the activation TopK is new. The paper is a good starting point as a review and a novel, if straight forward, combination of prior work. Simple techniques which work are valuable because of their simplicity and not in spite of it. It is difficult to achieve acceleration on existing hardware with this technique, but the authors provide lots of simulator / cycle analysis which could guide future hardware development. The experiments are rather extensive in supporting the efficiency of the method. There are some questions about the accuracy of the FLOP counts provided for some prior techniques (see Reviewer #2 re: RigL and SNFS). I hope the authors are able to correct this and other issues raised in the reviews in the final version.